# Comparative Genomic and Expression Analysis Insight into Evolutionary Characteristics of *PEBP* Genes in Cultivated Peanuts and Their Roles in Floral Induction

**DOI:** 10.3390/ijms232012429

**Published:** 2022-10-17

**Authors:** Chao Zhong, Zhao Li, Yunlian Cheng, Haina Zhang, Yu Liu, Xiaoguang Wang, Chunji Jiang, Xinhua Zhao, Shuli Zhao, Jing Wang, He Zhang, Xibo Liu, Haiqiu Yu

**Affiliations:** Peanut Research Institute, College of Agronomy, Shenyang Agricultural University, Shenyang 110000, China

**Keywords:** *PEBP*, gene family, flowering time, cultivated peanut

## Abstract

Phosphatidyl ethanolamine-binding proteins (PEBPs) are involved in regulating flowering time and various developmental processes. Functions and expression patterns in cultivated peanuts (*Arachis hypogaea* L.) remain unknown. In this study, 33 *PEBP* genes in cultivated peanuts were identified and divided into four subgroups: FT, TFL, MFT and FT-like. Gene structure analysis showed that orthologs from *A* and *B* genomes in cultivated peanuts had highly similar structures, but some orthologous genes have subgenomic dominance. Gene collinearity and phylogenetic analysis explain that some *PEBP* genes play key roles in evolution. *Cis*-element analysis revealed that *PEBP* genes are mainly regulated by hormones, light signals and stress-related pathways. Multiple *PEPB* genes had different expression patterns between early and late-flowering genotypes. Further detection of its response to temperature and photoperiod revealed that *PEBPs* *ArahyM2THPA*, *ArahyEM6VH3*, *Arahy4GAQ4U*, *ArahyIZ8FG5*, *ArahyG6F3P2*, *ArahyLUT2QN*, *ArahyDYRS20* and *ArahyBBG51B* were the key genes controlling the flowering response to different flowering time genotypes, photoperiods and temperature. This study laid the foundation for the functional study of the *PEBP* gene in cultivated peanuts and the adaptation of peanuts to different environments.

## 1. Introduction

Flowering is an important process that determines the success of plant reproduction. Plants must accurately combine internal and environmental signals to start the flowering process. The molecular mechanism of plant flowering is regulated by different pathways. The current research based on Arabidopsis has found that flowering pathways include light-dependent, autonomous, vernalization, gibberellin, temperature and age pathways [1,2]. Different pathways converge to downstream phosphatidylethanolamine-binding protein (*PEBP*) genes through molecular regulation, and these genes are transmitted downstream to floral meristem identify genes, which determine the formation of flowering [3].

PEBP is a kind of conserved protein found in plants that engages in several biological processes such as flowering time regulation and plant architecture management [1,2]. PEBP proteins are classified into three types: FLOWERING LOCUS T-like protein, TERMINAL FLOWER1 (TFL1)-like proteins and MOTHER OF FT AND TFL1 (MFT)-like proteins [4]. The FT-like subfamily in Arabidopsis has two members, FT and TWIN SISTER OF FT (TSF), both of which promote the transition from vegetative to reproductive growth [5]. FT is an activating protein that has been demonstrated to migrate from the leaves to the shoot apical meristem and interact with the transcription factor *FLOWERING LOCUS D* (*FD*) to speed up flowering [6]. *TSF*, the homologous gene of *FT*, promotes *A. thaliana* early flowering and has a redundant effect similar to that of *FT* overexpression [7]. *TFL1*, *Arabidopsis thaliana CENTRORADIALIS* (*ATC*) and *BROTHER OF FT* (*BFT*) are members of the *TFL1*-like subfamily, all of which have been shown to postpone flowering time [8,9]. *TFL1* has a role in inhibiting flowering and plant architectural control by regulating the meristem genes [10]. *ATC* is a short-day triggered floral inhibitor, and overexpression of *ATC* can enhance the late-flowering phenotype [4]. Another member of the *TFL1* subfamily is *FT’s* and *TFL1’s* brother (*BFT*). *BFT* overexpression in *A. thaliana* causes delayed flowering, although the *bft* mutant does not have the same phenotype as the *tfl1* mutant [11]. *MOTHER OF FT AND TFL1* (*MFT*) is the ancestor gene of *FT* and *TFL1* and is involved in seed germination and development. *MFT* is found in high concentrations in seeds and roots and induces seed dormancy [12]. Overexpression of *AtMFT* can result in early flowering, but not to the same extent as *FT*. Furthermore, *MFT* is expressed exclusively in seeds, mostly via the ABA and GA signaling pathways, to assist in seed germination control [13].

Flowering characteristics of crops are of great significance to crop production. As a core gene in the flowering process, the *PEBP* gene has been identified and analyzed in many corps, such as the Arabidopsis *FT* gene ortholog *Hd3a* in rice (*Oryza sativa*), *FT* homologs in soybean (*Glycine max*) and the *ZCN* genes in maize (*Zea mays*) [14,15,16]. *Hd3a*, a rice *FT* ortholog of Arabidopsis, moves from the leaf to the shoot apical meristem and induces flowering [16]. The *FT* homologous gene *GmFT1a* in soybean is a flower-forming inhibitor. When the condition changes from short-day to long-day, soybean varieties that are sensitive to photoperiods will have delayed flowering, and the *GmFT2a/5a* gene is a flower-promoting gene, which is opposite to the expression of *GmFT1a* [17,18]. *CENTRORADIALIS 8* (*ZCN8*) in maize (*Zea mays*) was expressed in the shoot apical meristem under short-day treatment, and its expression in the shoot apical meristem was not detected under long-day treatment. Therefore, *ZCN8* is a key gene for flower formation in maize [19].

The cultivated peanut (*Arachis hypogaea* L.) is an oil crop widely grown around the world, which not only provides abundant oil products but also contains rich nutrients. Flowering time has a strong impact on biomass production, and it is therefore necessary to understand how flowering is regulated in important crops such as peanut. The flowering time of peanuts directly determines the growth duration and is closely related to peanut yields and growing regions [20]. Investigating the molecular patterns of flowering genes in peanuts and improving the effective flowering time can increase peanut production [21,22]. As an allotetraploid plant, peanut (AABB, 2n = 4x = 40) is produced from two diploid species, *A. duranensis* (AA genome) and *A. ipaensis* (BB genome) [23,24]. The *PEBP* genes in diploid wild peanut species were identified, and the *PEBP* genes in cultivated peanuts and tissue-specific expression pattern based on transcriptome data was analyzed [25]. In order to further understand the gene structure and function of *PEBP* genes in cultivated peanuts, in the present study, we re-identified the *PEBP* genes of cultivated and wild peanuts, explored their evolutionary characteristics in cultivated peanuts through comparative genomics, and further identified their floral induction and mechanisms to genoytpes, photoperiod and temperature. This study will contribute to further insight into the molecular function and regulation of *PEBP* genes in peanuts.

## 2. Results

### 2.1. Identification of PEBP Proteins

The Hidden Markov Model PF01161 was used as a query to identify the *PEPB* genes in the peanut genome and its two ancestors, and total of 64 protein candidate sequences have been identified in three *Arachis* genomes including 16 of *A. ipaensis*, 16 of *A. duranensis* and 32 of *A. hypogaea*. Six PEBP proteins of *A. thaliana* were used to perform a genome-wide search on 3 *Arachis* species, and all 64 genes were confirmed to be homologous with *Arabidopsis* (Appendix A). These protein sequences were submitted to databases CDD, Pfam and SMART for identification of their PEBP conserved domain, and all the genes contained complete PEBP domains. A total of 32 protein sequences from *A. hypogaea*, 16 from the *A. duranensis* and 16 from the *A. ipaensis* were identified as PEBP proteins. The PEBP protein sequences of peanuts and *Arabidopsis thaliana* were clustered and compared with the gene structure, and it was found that *PEBP* genes were divided into 4 different types including FT, TFL, MFT and FT-like (Figure 1, Appendix A). Among the 33 FT-type genes, there are 16 *A. hypogaea* genes, 8 *A. duranensis* genes and 9 *A. ipaensis* genes, while the TFL-type genes contains 8 *A. hypogaea* genes and 8 wild peanut genes. Among the 8 MFT-type genes, there are 4 wild peanut genes and 4 cultivated peanut genes, and the FT-like contains 4 cultivated peanut and 3 wild peanut genes. The full length of the candidate PEBP protein sequence, the molecular weight (MW), the isoelectric point (pI) and predicted subcellular localization were summarized (Appendix A). The full length of the 53 protein sequences ranges from 103 to 216. The MW (kDa) range of these protein sequences is from 11.20 to 24.70, and pI is from 4.85 to 9.68. The vast majority of genes were predicted to be localized in the cytoplasm. All parameters are consistent with the range of the PEBP domain.

### 2.2. Structure and Orthologous Relationship Analysis of PEBP Genes

Conserved motifs in the 64 PEBP proteins were identified using the MEME online tool. In total, 15 motifs were identified in the peanut proteins, and the motifs identified were 6 to 50 amino acids in length (Figure 1 and Appendix A). The FT, TFL and MFT genes contain similar motifs, but no motif is shared by FT, TFL and MFT, and the motifs in FT-like are different to the other three classes, implying that it may be functionally different from typical *PEBP* genes. When comparing the PEPB proteins of cultivated peanuts with wild peanut, the genes in *A. hypogaea* are differentiated in structure; for example, motif 1 is absent in wild peanut protein *AraduA6WCN* and *Arip03WUR* compared with their orthologies, *ArahyG6F3P2* and *Arahy4GAQ4U,* in cultivated peanuts. In TFL sub-family, *Arahy02PX6U* and *AraHyCIFX90* contains motif 15 while their orthologies, *Araip4V81G* and *Araip344D4,* do not. The intron-exon structure pattern was explored based on the genome annotation file (GFF) to acquire a better understanding of the structural diversity of *PEBP* genes. Compared with the diploid wild peanut, *A. hypogaea* has four typical exons and three introns, except for FT-like, and has similar UTR and intron length, while the wild peanut has nine genes that did not fit this profile.

Orthologous relationships were identified between the A and B subgenomes of *A. hypogaea* and the wild diploid peanut based on phylogenetic analysis for FT, TFL and MFT clades (Figure 2). The results showed that *PEBPs* had high similarity between orthologous genes, indicating that *PEBP* genes were highly conserved in the evolutionary process. The similarity between A and B subgenomes reached more than 90%, except for the gene pair *ArahyIZ8FG5* and *ArahyE8V3SA,* which were only 57.39% similar. The similarity of five genes between the two wild diploids was less than 80%, indicating that the sequences of the A and B subgenomes of the cultivated peanut tended to be more similar when compared to the pairs between *A. duranensis* and *A. ipaensis*. Some genes retained high similarity to *A. duranensis*, such as *ArahyLUT2QN* and *ArahyIZ8FG5*, and some genes retained high similarity to *A. ipaensis*, such as *Arahy9K247M, Arahy2I11L4* and *ArahyPBX2T9*. Interestingly, some genes in the A and B subgenomes produced higher similarity to each other than their wild ancestor, such as gene pairs *ArahyG6F3P2* and *Arahy4GAQ4U*, *ArahyS8RL65* and *ArahyMKP2FL*, *ArahyFFIH3X* and *ArahyNBC3HK*, *Arahy02PX6U* and *ArahyCIFX90* and ArahyDYRS20 and ArahyBBG51B. The sequence variation characteristics of different genes in cultivated peanuts may be the result of the adaptation of flowering traits to the environment.

### 2.3. Distribution on Chromosomes and Synteny Analysis of PEBP Genes

The distribution map of the *PEBP* genes was drawn based on the gene position on the chromosome (Figure 3). The 32 *PEBP* genes of cultivated peanuts were distributed on 16 chromosomes and corresponded to those of wild peanuts. The chromosomal location of *PEBPs* in cultivated peanuts is consistent with the majority *PEBPs* of wild peanuts. Combined with the results of orthologous analysis, *AraduEHZ9Y* was absent in the corresponding position of cultivated peanuts. *AraipYA5YU* was not found in the corresponding position of chromosome 15 in the B subgenome of cultivated peanuts, but appeared on chromosome 5 in A subgenome, which may suggest that the gene has undergone homologous recombination between the A and B subgenomes. The location of *ArduG0NJW* is inconsistent with its orthologous gene, *ArahyM2THPA*, probably because *ArduG0NJW* is transposed in wild peanuts. This result showed that the peanut *PEBP* genes were homologous to that of wild peanuts, but some genes were transposed or recombined.

To further explore the evolutionary relationship of *PEBP* genes between cultivated peanuts, *A. hypogaea**,* and other species, collinearity was investigated between *PEBPs* and homologs among other species. Homolog pairs were identified between *PEBPs* in *A. hypogaea* and other plants, including wild diploid peanuts, soybean, chickpea, alfalfa, common beans, Arabidopsis, grapes, rice, sorghum and maize (Figure 4). The number of colinear pairs between *A. hypogaea* and wild peanut species was both 28, respectively. The number of gene pairs in cultivated peanuts and other legumes including soybean, common bean, alfalfa and chickpea is 41, 22, 21 and 18, respectively; the number of gene pairs in the dicotyledonous plants grape and Arabidopsis is 20 and 6, and the gene pairs of monocotyledonous plants sorghum, rice and maize are 20, 18 and 12, respectively. The results show that the genetic relationship between *PEBPs* in cultivated peanuts and *PEBPs* in soybean, wild peanuts, common bean and alfalfa is close, and has a long distance of phylogenetic relationship with Arabidopsis. The *TFL* genes, *Arahy02PX6U*, *ArahyCIFX90*, *ArahySB1I92*, *Arahy9K247M, ArahyDYRS20* and *ArahyBBG51B* exhibit the most homologous gene pairs with other species (Appendix A). These 6 genes were present in a total of 22 to 25 gene pairs in cultivated peanuts and other species, compared with only 2 to 9 gene pairs for other PEBP genes. These results suggest these genes may play crucial roles in the *PEBP* gene family during evolution and have important functions.

The Ka/Ks ratios of *PEBP* gene pairs were determined to better understand the evolutionary limitations impacting the *PEBP* gene family. The Ka/Ks ratio of gene pairings between *PEBP* genes and their orthologous genes was less than 1, suggesting that the *PEBP* gene family has been subjected to intense selection pressure during evolution (Appendix A).

### 2.4. Phylogenetic Analysis for PEBP Genes among Cultivated Peanust and Other Plant Species

To investigate the phylogenetic relationship between the *PEBP* genes in the *Arachis* and other plants, a phylogenetic tree was constructed with the Neighbor-joining method (Figure 5). In total, 188 PEBP protein sequences in wild diploid peanuts, soybean, chickpea, alfalfa, common beans, Arabidopsis, grapes, rice, sorghum and maize were used for multiple sequence alignment to construct a phylogenetic tree. The PEBP sequences of *A. ipaensis*, *A. duranensis* and *A. hypogaea* were classified into four different subgroups, MFT, TFL, FT and FT-like subgroups, respectively. In the four subgroups, the number of *PEBP* genes in the MFT subgroup is 23. In the TFL subgroup, there are a total of 54 genes, and the FT subgroup contains a total of 95 genes. Most of the two genes in the two diploid *Arachis* are orthologous to the two genes in the cultivated peanut, *A. hypogaea*, which is consistent with the distribution of genes on the chromosomes (Figure 3). The phylogenetic analysis showed that the *PEBP* genes of peanuts have a close genetic distance with other legumes.

### 2.5. Cis-Element in the Promoter Region of PEBP Genes

Promoter activities play a crucial role in regulating gene functions. To understand the genetic functions, metabolic networks, and regulatory mechanisms of *PEBP* genes, the *cis*-elements in their promoter regions are analyzed. In total, 1648 cis-acting elements of 51 types were predicted to contain potential functions (Figure 6). The different types of *cis*-elements were classified into 14 categories according to the type of function-related and regulatory pathways (Figure 6), and the homologous genes from A and B genomes, respectively, which had similar promoter elements. These *cis*- elements are functionally involved in growth and development, hormonal regulation and response to stress. Among the 14 different functional categories, the most abundant are 853 light response-related *cis*-elements, which also indicate that flowering regulation in peanuts is closely related to light response. The next is hormone regulation-related elements, including 177 MeJA, 180 Abscisic acid, 73 Gibberellin and 39 Auxin responsiveness-related elements, indicating that flowering regulation in peanuts may depend on these hormone regulations. In addition, there are some *cis*-acting elements related to stress response, such as drought and low temperature.

### 2.6. Phenotypic Identification of Peanut Cultivars with Different Flowering Times in Response to Photoperiod and Temperature

Four peanut genotypes with different flowering times were selected to identify responses to temperature and photoperiod, namely the early-flowering genotypes Silihong (abbreviation: SLH) and Baihua5 (abbreviation: BH5), and the late-flowering genotypes Huayu22 (abbreviation: HY22) and Tifrunner (abbreviation: TIF). The early and late-flowering genotypes were placed in three environments, long day and high temperatures (LDHT), short day and high temperatures (SDHT) and short day and low temperatures (SDLT). The results showed that there was no significant difference in the number of total flowers and the length of the flowering period. The flower initiation time and the whole flowering period showed significant delays under different temperature treatments but did not change significantly between photoperiod treatments (Figure 7, Appendix A). Under the same environmental treatment, the early-flowering genotypes SLH and BH5 started flowering 9–11 days earlier than the late-flowering genotypes TIF and HY22. Compared with SDHT treatment, the flowering time of the four genotypes did not change significantly under the long-day treatment (LDHT), while the flowering time of the four genotypes under low temperature (SDLT) was delayed by 4–5 days, respectively (Figure 7).

### 2.7. PEBP Expression Pattern Analysis Responds to Genotypes, Temperature and Photoperiod in Early and Late Flowering Genotypes

To elucidate the involvement of *PEBP* genes causing differences in flowering time, the early and the late-flowering genotypes were selected for expression analysis of transcripts (Figure 8). Genes with a two-fold difference in gene expression levels between early-flowering and late-flowering genotypes at a time point were considered to have differences in expression patterns. Some genes were expressed at lower levels and showed no significant difference in expression patterns between early and late flowering genotypes, including *Arahy**V0UB6N*, *Arahy**L2SG6N*, *Arahy**S8RL65*, *Arahy**MKP2FL*, *Arahy**VQ1Q3Q*, *Arahy**E8V3SA*, *Arahy**5H2LSK*, *Arahy**T6PK46*, *Arahy**FFIH3X*, *Arahy**NBC3HK*, *Arahy**SB1I92*, *Arahy**9K247M*, *Arahy**PBX2T9* and *Arahy**TCE5WS*. It indicates that these genes may not have the function of regulating flowering time in cultivated peanuts. Some orthologies showed similar expression patterns between A and B subgenomes and significant differences in expression patterns between early-flowering and late-flowering genotypes, including *Arahy**XGVA1E* and *Arahy**JQBA31*, *Arahy**M2THPA* and *Arahy**EM6VH3*, *Arahy**DYRS20* and *Arahy**BBG51B* and *Arahy**02PPX6U* and *Arahy**CIFX90*. It suggested that these genes may be involved in the regulation of flowering time in cultivated peanuts. Interestingly, some genes showed different expression patterns between A and B subgenomes, including *Arahy**LUT2QN* and *Arahy**VQ1Q3Q*, *Arahy**IZ8FG5* and *Arahy**E8V3SA* and *Arahy**2I11L4* and *Arahy**VGT3DW*, which revealed that the mechanism of *PEBP* genes regulation of flowering time is different between the A and B subgenomes of cultivated peanuts.

The initial flowering time of early and late-flowering genotypes was at 32 d and 42 d under SDHT treatment (Figure 7), respectively, so they were chosen to examine expression level to different photoperiods and temperatures (Figure 9). Although the flowering time of the four genotypes did not change significantly under different photoperiod treatments, however, the expression levels of some genes changed under long-day treatment, a total of 10 genes showed significant differences in expression and 5 genes, *Arahy4GAQ4U*, *ArahyDYRS20*, *ArahyEM6VH3*, *ArahyG6F3P2* and *ArahyLUT2QN,* showed different expression levels in all the 4 genotypes. These genes showed the same change pattern under different photoperiods. For example, *ArahyLUT2QN* and *ArahyG6F3P2* in the FT subfamily were up-regulated under long-day treatment, while *ArahyM2THPA*, *Arahy4GAQ4U*, *ArahyEM6VH3* and *ArahyIZ8FG5* were down-regulated. In the TFL gene family, *ArahyDYRS20* and *ArahyBBG51B* was down-regulated. It indicates that these genes play a role in responding to different photoperiods in cultivated peanuts. A total of 12 genes showed significant differential expression under low temperature treatment, among which 6 genes, *ArahyBBG51B*, *ArahyDYRS20*, *ArahyG6F3P2*, *ArahyEM6VH3*, *ArahyLUT2QN* and *ArahyM2THPA*, showed differences in expression between early-flowering and late-flowering genotypes. The expression levels of *ArahyLUT2Q* and *ArahyG6F3P2* were up-regulated under low temperature treatment, while other genes were down-regulated, indicating that these genes are involved in the regulation mechanism of peanut flowering in response to environmental temperature. These suggested that the elevated expression of *ArahyM2THPA*, *ArahyEM6VH3*, *Arahy4GAQ4U* and *ArahyIZ8FG5* in the FT gene family can promote flowering while *ArahyG6F3P2* and *ArahyLUT2QN* inhibits flowering. The elevated expression of *ArahyDYRS20* and *ArahyBBG51B* in the TFL gene family can promote flowering.

## 3. Discussion

The peanut is a globally important oil crop, and the identification of the peanut *PEBP* gene contributes to providing insights into the regulation of flowering time in peanuts. As an allotetraploid crop, cultivated peanuts, *A. hypogaea,* were derived from the hybridization between the diploids *A. duranensis* (A genome) and *A. ipaensis* (B genome) [26,27,28]. Although the *PEBP* gene of cultivated peanuts was searched together with wild peanut, the *PEBPs’* function in cultivated peanuts remains unclear [25]. The *A. hypogaea* genome has evolved through homeologous recombination between A and B subgenomes, and the cultivated peanut has dramatic differences in agronomic traits from their wild ancestors [26,28]. Exploring evolutionary features and expression profiles of *PEBPs* genes in cultivated peanuts will be beneficial for further understanding the functions of *PEBPs* in peanuts. Therefore, this study aimed to identify the *PEBP* gene family in cultivated peanuts, *A. hypogaea*, which was sequenced recently, to directly explore the evolutionary characteristics of *PEBP* genes, as well as the expression patterns response to genotypes, temperatures and photoperiods in controlling flowering time.

The two diploid wild species of peanuts are believed to have been hybridized and polyploidized to create the allotetraploid cultivated peanut [25,27,28,29]. In this study, each of the diploid wild peanuts contains 16 PEBP genes, which was confirmed by Jin et al. in 2019; however, a total of 32 *PEBP* genes identified in this study are inconsistent with the number of previous studies [25]. It may be due to the different parameters we set to identify *PEBP* proteins. Since *Arahy**FW8Z6T* contains only 78 amino acid residues, it is much shorter than the amino acid length of a typical PEBP protein and could not be identified by HMM-based methods. Therefore, we excluded *Arahy**FW8Z6T*. *Arahy**1WC97K* and *Arahy**T6PK46*, which were not included in Jin’s (2019) study, in gene length, isoelectric point and conserved domains, all fit the parameters of PEBP protein and were therefore retained (Appendix A). *PEBP* proteins in cultivated peanuts can be divided into four different subfamilies: FT, TFL, MFT and FT-like (Figure 1). *PEBP* genes in angiosperm including wild diploid peanuts are identified as three groups, including FT, TFL and MFT [29]. In this study, two *A. duranensis* genes, one *A. ipaensis* gene and four *A**. hypogaea* genes were classified as FT-like. These genes were found to have distinct conserved motifs content compared with other FTs, TFLs and MFTs through identification of conserved domains. Further sequence analysis showed that they were closer to prokaryotes [30]. The *PEBP* genes, which are atypical in eukaryotes, are also present in other plants including wheat, cotton and potato [31,32,33]. Whether these genes are involved in flowering and other biological functions of cultivated peanuts requires further study. An alignment of homologous genes from cultivated peanut A and B subgenomes with those of wild peanuts revealed that the similarity between A and B subgenomes of cultivated peanut was higher than that between the two wild peanut species, probably due to extensive homologous recombination between the A and B subgenomes [27]. In addition, different *PEBP* genes have subgenomic dominance between the A and B genomes. For example, *Arahy**LUT2QN* and *Arahy**IZX8FG5*, which are highly homologous to *A.*
*duranensis*, have significant changes in response to different flowering-time genotypes, photoperiods and temperatures, while the homologous genes *Arahy**JQBA31* and *Arahy**E8V3SA* from the B subgenome did not show significant expression changes. These results show that the *PEBP* genes of cultivated peanuts have a high sequence similarity with that of wild peanuts, however, there may be novel mechanisms from wild peanuts in terms of gene sequence structure and expression pattern.

The initial flowering date of peanuts directly determines their growth duration, which is closely related to peanut yield and planting region, and photoperiod and temperature are two important factors that affect flowering. The *PEBP* gene’s response to genotypes, photoperiod and temperature was characterized in this study. Integrating the expression patterns of the *PEBP* gene under different treatments found that the MFT subfamily had no significant changes in expression patterns under different genotypes, photoperiods and temperature treatments, and only *Arahy2I11L4* detected a higher expression level in the late flowering genotypes than early flowering genotypes. It indicated that *MFT*s may not have a role in peanut flowering induction. Although *MFT* genes have been shown to be involved in flowering regulation in Arabidopsis, the function of *MFT* genes is mainly involved in the development of plant seeds [9,34,35,36]. In the transcriptomes of 22 cultivated peanut tissues, *MFT*-type genes were found to be significantly increased in pod tissues, which also indicated that *MFT* genes were involved in the development of peanut pods in cultivated peanuts [25,37]. The *MFT* promoter region of cultivated peanuts contains more ABRE cis-elements that respond to ABA regulatory pathways. This indicates that the *MFT* gene may regulate the seed development of cultivated peanuts through the ABA signaling pathway [38].

Six *TFL* genes, *Arahy02PX6U*, *ArahyCIFX90*, *ArahySB1I92*, *Arahy9K247M, ArahyDYRS20* and *ArahyBBG51B* exhibit the most homologous gene pairs with other species identified through synteny analysis. Although *Arahy02PX6U*, *ArahyCIFX90*, *ArahySB1I92* and *Arahy9K247M* did not show significant expression differences among genotypes with different flowering time and under different photoperiod and temperature treatments, these genes may be involved in other biological processes of peanut growth and development. *TFL* genes not only control flowering but also play a key role in plant architecture. *AraduF950M*, *AraduWY2NX*, *Araip344D4* and *Araip4V81G* are predicted to control wild peanut plant architecture and are orthologous genes of *Arahy02PX6U*, *ArahyCIFX90*, *ArahyDYRS20* and *ArahyBBG51B* [25]. *ArahyBBG51B* has been shown to be a key gene controlling peanut flowering patterns, and it accumulates on the main stem of alternate flowering type peanut to repress flowering [39]. In this study, low temperature inhibited flowering, and the expression of *ArahyBBG51B* and its ortholog *ArahyDYRS20* decreased under long-day and low-temperature treatments, which may be because the distribution of gene transcripts in leaves was different from that of flowering tips.

In the FT subfamily, the expression level of *FT* genes in the early-flowering genotype was higher than that in the late-flowering genotype, which may mean that their increased expression promotes flowering. However, under SDLT and LDHT conditions, the expression levels of *ArahyLUT2QN* and *ArahyG6F3P2* were increased, while the expression levels of *ArahyM2THPA*, *ArahyEM6VH3*, *Arahy4GAQ4U* and *ArahyIZ8FG5* were lower than those of SDHT. It indicates that the *FT* genes may have opposite functions under different photoperiod and temperature treatments, which has been confirmed in soybeans [17,18,40,41]. The increased expression of *GmFT2a* and *GmFT5a* promoted flowering under short-day and high-temperature conditions [17,18,42]. *ArahyM2THPA* and *ArahyEM6VH3* are located on the same branch as *GmFT5a* in phylogenetic analyses, and their responses to photoperiod and temperature are similar to *GmFT5a*, while *ArahyIZ8FG5* and *GmFT2a* are closer in phylogenetic analyses, and both genes respond to photoperiods. *ArahyLUT2QN* and *ArahyG6F3P2* lie in the same phylogenetic clade as *GmFT3a/3b* and *GmFT4*, respectively, and elevated *ArahyLUT2QN* inhibits flowering, and *GmFT3a/3b* was shown to respond to long-day light, consistent with the expression patterns of *ArahyLUT2QN* and *ArahyG6F3P2*. In addition, collinearity analysis of multiple species found that cultivated peanuts and soybeans have the most collinear gene pairs, indicating that the *PEBP* gene of the cultivated peanut has the closest genetic distance to soybeans, which was also confirmed in a phylogenetic analysis. It means that the *PEBP* genes in peanuts may have high homology and similar functions to those in soybeans. 

The four genotypes with different flowering times did not show early or delayed flowering under long days, which may be that the genotypes used in this study were insensitive to photoperiods. A previous study revealed that photoperiods had little influence on the time to flowering in peanuts but affects their reproductive development in many ways by influencing the processes that occur mainly after flowering [22,43]. The flowering time of peanuts is sensitive to temperature [44]. In this study, four peanut genotypes showed delayed flowering time under low-temperature treatments. Compared with photoperiod treatments, more *PEBP* genes showed significant changes in low-temperature treatments, indicating that cultivated peanuts are more sensitive to temperature than photoperiod. Regardless of different photoperiods, temperatures or genotypes with different flowering times, *ArahyM2THPA*, *ArahyEM6VH3*, *Arahy4GAQ4U*, *ArahyIZ8FG5, ArahyG6F3P2*, *ArahyLUT2QN, ArahyDYRS20* and *ArahyBBG51B* showed significant changes in expression level, indicating that they are the key genes controlling flowering in cultivated peanuts. However, the flowering induction of plants in response to different genotypes, temperatures and photoperiods is a complex regulatory network, and the functions of all *PEBP*s require in-depth functional studies of individual genes to explain their environmental and genetic molecular mechanisms.

## 4. Materials and Methods

### 4.1. Plant Materials and Growth Conditions

All plant materials were provided by the Peanut Research Institute of Shenyang Agricultural University. All seeds were sown in pots and transferred to an artificial climate chamber (PGC-40L2, Percival, Hong Kong, China) with a photosynthetic photon flux density of 700 µmol m^−2^ s^−1^ and relative humidity of 30%, 20 days after emergence.

Three different environmental conditions were set, and long days and high temperatures (LDHT) were represented by 16 h 30 °C light/8 h 25 °C dark cycles, short days and high temperatures (SDHT) by 10 h 30 °C light/14 h 25 °C dark cycles and short days and low temperatures (SDLT) by 10 h 25 °C light/14 h 20 °C dark cycles. The flowering time of each flower of different genotypes is recorded.

### 4.2. Download for Genome Data

Genomic data of cultivated peanuts and their two ancestors were obtained at the website PeanutBase (https://peanutbase.org/ (accessed on 1 April 2022)), including genome sequence, annotation and transcript files. The whole-genome data including DNA, CDS, protein sequence and genome annotation files of Arabidopsis, rice and soybean were downloaded from the Phytozome database (https://phytozome-next.jgi.doe.gov/ (accessed on 5 April 2022)).

### 4.3. Genome-Wide Identification of PEBP Proteins

To identify the target genes, we used two methods to identify members of the *PEBP* gene family. Firstly, the Hidden Markov Model PF01161 of PEBP domain from the Pfam database (https://pfam.xfam.org (accessed on 15 April 2022)) was downloaded and searched in the peanut protein database as a query using the Simple HMM Search module in TBtools software [27]. The genes with E-value < 10^−3^ were selected as the candidate *PEBP* proteins. The second method is to apply the BLAST search for PEBPs in the peanut protein database using the amino acid sequence of PEBP proteins AtFT (AT1G65480), AtTSF (AT4G20370.1), AtBFT (AT5G62040.1), AtTFL1 (AT5G03840.1), AtATC (AT2G27550.1) and AtMFT (AT1G18100.1) in Arabidopsis as queries, and then the newly searched proteins in which E-values < 10^−5^ are used to perform a BLAST search again to determine all the orthologs. To further confirm PEBP proteins, all the identified proteins were submitted to databases CDD (https://www.ncbi.nlm.nih.gov/cdd (accessed on 25 April 2022)), Pfam (http://pfam.xfam.org/ (accessed on 1 May 2022)) and SMART (http://smart.embl-heidelberg.de/ (accessed on 1 May 2022)) to verify whether they contained complete PEBP domain. Molecular weight (MW) and theoretical isoelectric point (PI) for the obtained PEBP protein sequences were calculated using ExPASy (https://web.expasy.org/protparam/ (accessed on 5 May 2022)). The same process above was applied to the diploid ancestors *A. duranensis* and *A. ipaensis*.

### 4.4. Chromosome Location, Phylogenetic Analysis

The distribution map of *PEBP* family genes of cultivated peanut and its two ancestors on each chromosome was constructed with TBtools based on genome annotation files (GTF/GFF). To further understand the phylogenetic relationship of PEBP proteins in three *Arachis* species and other plant species, the phylogenetic tree was constructed using all PEBP protein sequences obtained from wild diploid peanuts, soybean, chickpea, alfalfa, common beans, Arabidopsis, grapes, rice, sorghum and maize. The PEBP protein sequences were aligned using the MUSCLE algorithm in MEGA-X (https://www.megasoftware.net/ (accessed on 5 May 2022)), and the neighbor-joining (NJ) tree was built with the bootstrap value set at 1000 repetitions [45].

### 4.5. Gene Structure and Conserved Motifs Characterization of PEBP Genes

The TBtools software was used to analyze the gene structure of the *PEBP* genes on chromosomes based on the genome annotation files (GTF/GFF) of peanuts and their diploid ancestors, *A. duranensis* and *A. ipaensis*. The MEME program (http://meme-suite.org/ (accessed on 15 May 2022)) was used to detect conserved motifs of the PEBP family proteins with the following parameters: the width of the motif ranged from 6 to 50 amino acids, with a maximum of 20 [46]. Only motifs with an E-value < 10^−10^ were retained for further analysis. Other options used the default values. Gene cluster analysis, gene structure and conserved domain maps were integrated using TBtools.

### 4.6. Gene Collinearity and Selection Pressure Analysis

The MCScanX was used to examine duplication genes [47]. The *PEBP* homoeologs between cultivated peanuts and its two diploid ancestors were analyzed. To investigate selection pressure, we estimated the non-synonymous to synonymous mutation rate (Ka/Ks) of homologous genes. The Ka (non-synonymous substitution rate) and Ks (synonymous substitution rate) values of repetitive genes were calculated using the Ka/Ks Calculator 2.0 program to determine selection pressure [48].

### 4.7. Cis-Element Analysis of the PEBP Genes

We extracted the 2000 bp upstream area of the start codon (ATG) and submitted the sequences to the PlantCARE (http://bioinformatics.psb.ugent.be/webtools/plantcare/html/ (accessed on 20 May 2022)) database to see if the cis-regulatory model of *PEBP* genes was conserved [49].

### 4.8. Quantitative RT-PCR Analysis for Gene Expression

To study the expression patterns of *PEBP* gene in early and late flowering genotypes, mature leaves of peanut plants were collected at 20 d, 25 d, 30 d, 35 d and 40 d after sowing. For studies of growth temperature and photoperiod, mature leaves under three environmental conditions (LDHT, SDHT and SDLT) were collected at two key flowering periods, 32 d and 42 d after sowing. All leaf tissues were collected 2 h after the light was turned on and stored at −80 °C for detection of *PEBP* gene expression.

Total plant RNA was extracted using the Plant Total RNA Extraction Kit (Centrifugal Column) (Tiangen Biotech, Beijing, China). cDNA was synthesized using the Reverse Transcription Kit PrimeScriptTM RT Reagent Kit (TaKaRa, Dalian China), and the procedure was based on its instructions. Primer design for detecting differential expression of *PEBP* genes was performed using PrimerBlast (https://www.ncbi.nlm.nih.gov/tools/primer-blast/ (accessed on 25 May 2022)) (Appendix A). The *Actin11* gene was used as the internal control. The reaction system for gene expression analysis was prepared with the kit SYBR Premix Ex TaqII (TliRNaseH Plus) (TaKaRa, Dalian China), and the instrument for fluorescence quantitative reaction detection was ABI 7500 (Applied Biosystems, Shanghai China). Relative expression analysis was calculated using the 2^−ΔΔCT^ method [50].

## Figures and Tables

**Figure 1 ijms-23-12429-f001:**
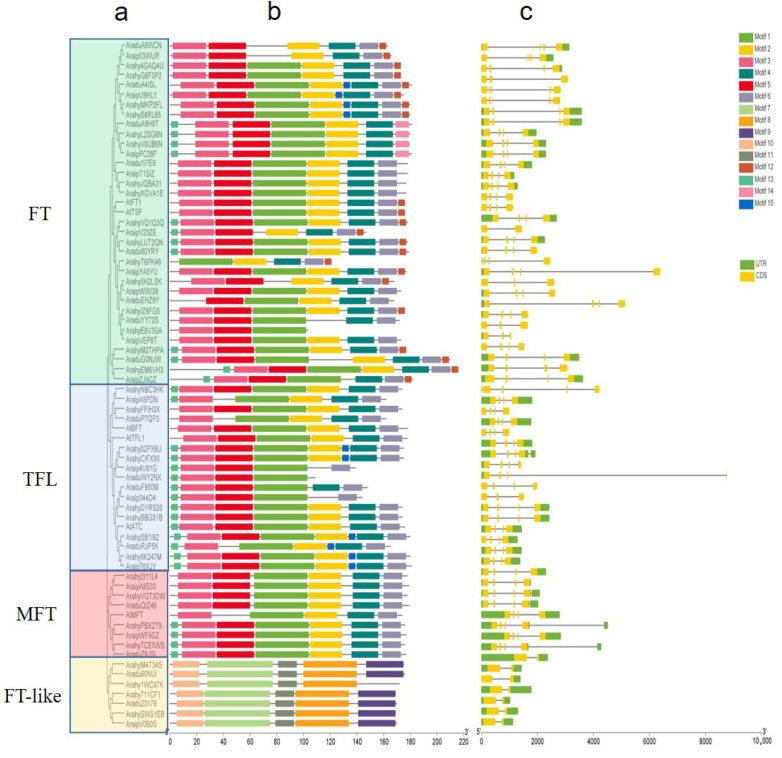
The phylogenetic tree, motifs and gene intron/exon structure of 64 PEBP genes in cultivated peanuts. (**a**) Phylogenetic tree based on PEBP protein sequence using Neigbour-joining Tree method. (**b**) The motif compositions of PEBP proteins. Specific motifs were indicated using different colors. (**c**) The exon and intron distribution of PEBP genes. Exons and intron regions are represented by yellow rectangles and grey lines, respectively.

**Figure 2 ijms-23-12429-f002:**
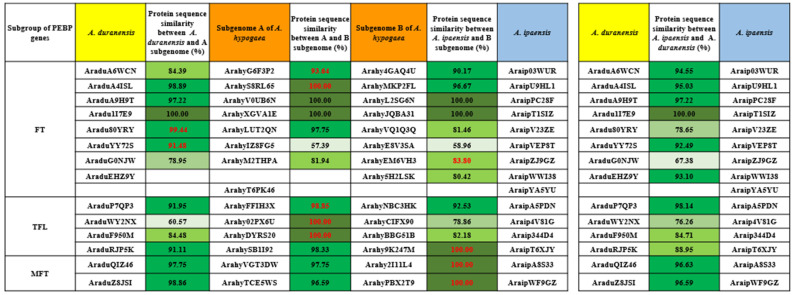
Orthologous gene identification and its protein sequence-based similarity analysis. Three different peanut genomes are marked with different background colors. Yellow, blue and orange represent the genomes of diploid wild peanut A. duranensis and A. ipaensis and tetraploid cultivated peanut A. hypogaea, respectively. Protein sequences from high to low similarity are marked with different depths of green background. The red font represents a pair with high similarity between orthologous genes.

**Figure 3 ijms-23-12429-f003:**
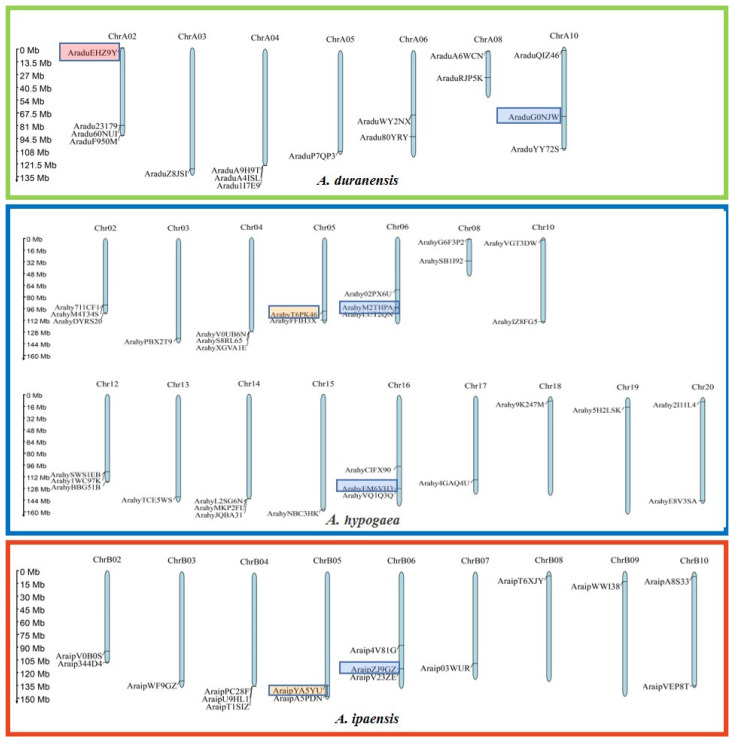
Chromosomal distribution of PEBP genes in peanut genome and its two ancestors. The green, red and blue rectangles represent the three *Arachis* genomes of two diploid *A**. ipaensis, A. duranensis* and the tetraploid *A. hypogaea*, respectively. The blue bars represent the chromosomes, and the *PEBP* genes were marked on different chromosomes. Red frame means that gene is absent in the corresponding position of cultivated peanuts. Yellow frames mean that genes have undergone homologous recombination between the A and B subgenomes. Blue frames mean that genes have transposed in wild peanuts.

**Figure 4 ijms-23-12429-f004:**
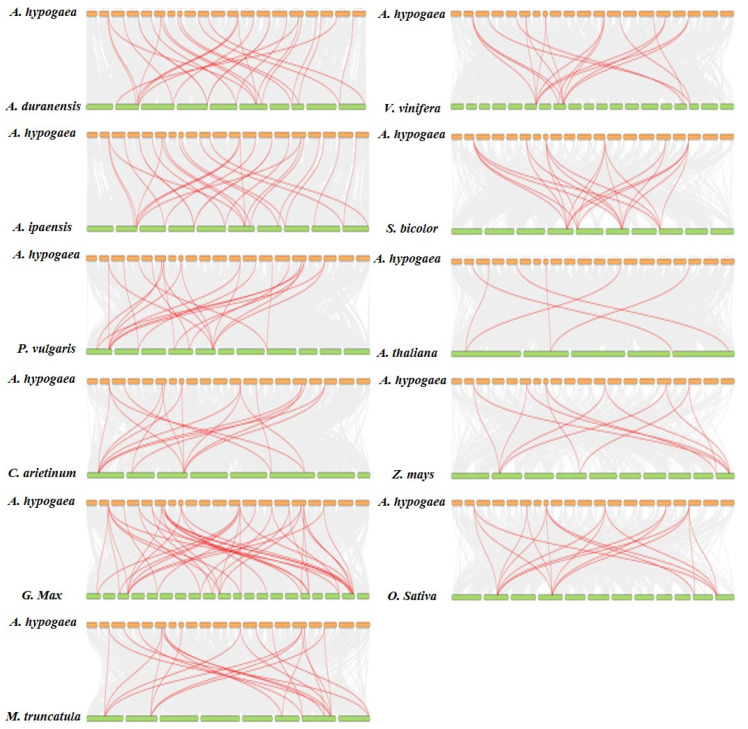
Syntenic analysis of PEBP genes between cultivated peanuts and other plants, including wild diploid peanuts, soybean, chickpea, alfalfa, common beans, Arabidopsis, grapes, rice, sorghum and maize. The gray lines at the bottom represent the collinear regions in rice and other genomes. The red lines represent the pairs of PEBP genes.

**Figure 5 ijms-23-12429-f005:**
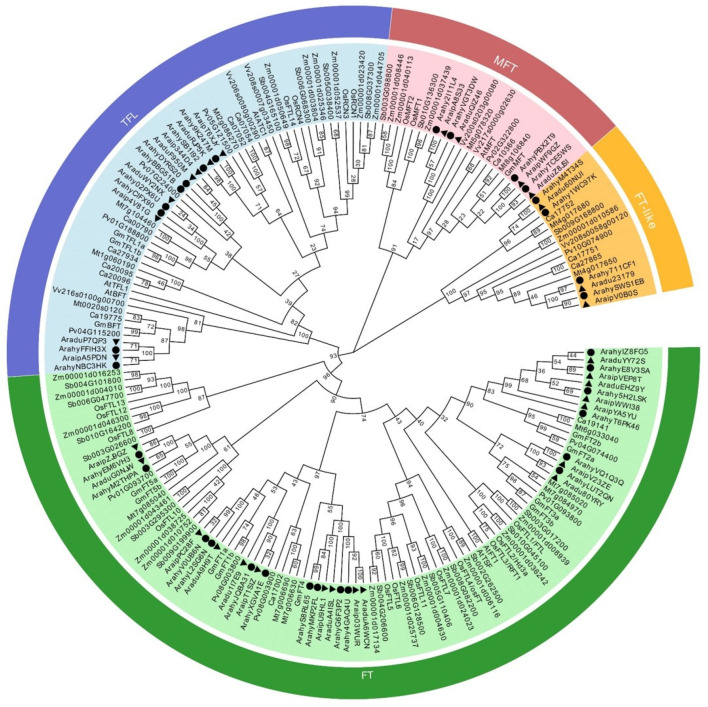
Neighbor-joining phylogenetic tree constructed based on 188 PEBP proteins of soybean, chickpea, alfalfa, common beans, Arabidopsis, grapes, rice, sorghum and maize as well as three Arachis species. The PEBP genes were divided MFT, FT and TFL and FT-like subgroups. PEBP proteins in cultivated and wild peanuts are marked by black dots and triangles, respectively. The phylogenetic tree is built using the complete amino acid sequences of the PEBP proteins by MEGAX with the Neighbor-joining method (Bootstrap = 1000).

**Figure 6 ijms-23-12429-f006:**
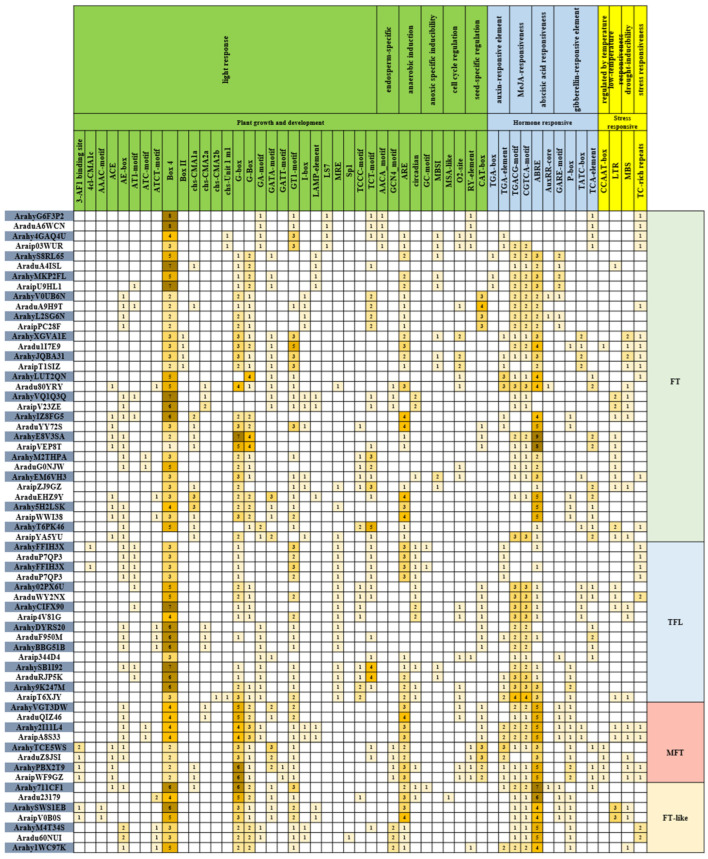
Analysis of cis-acting elements based on 2 kb upstream of PEBP genes. The number of cis-acting elements with various functions were counted. The number of cis-acting elements from 1 to 9 are marked with different depths of yellow back-ground.

**Figure 7 ijms-23-12429-f007:**
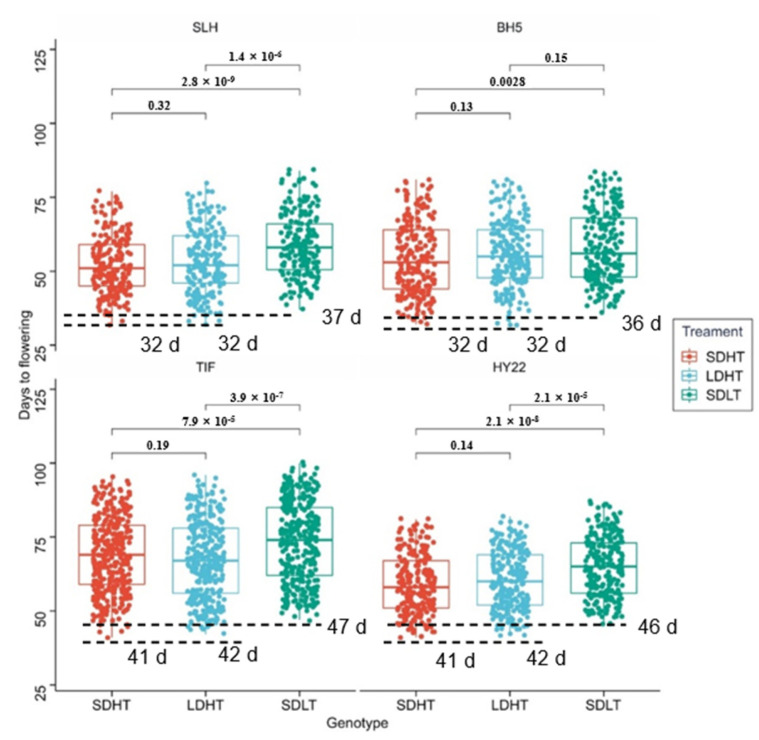
Responses of early and late-flowering genotypes under different temperature and photoperiod treatments; the horizontal axis represents the treatments, respectively, the vertical axis represents time (days) and each point represents the flowering time of each flower. The Kruskal-Wallis test was used to compare the distribution of flowering period under different treatments. Three different environmental conditions were set, long days and high temperatures (LDHT), short days and high temperatures (SDHT) and short days and low temperatures (SDLT).

**Figure 8 ijms-23-12429-f008:**
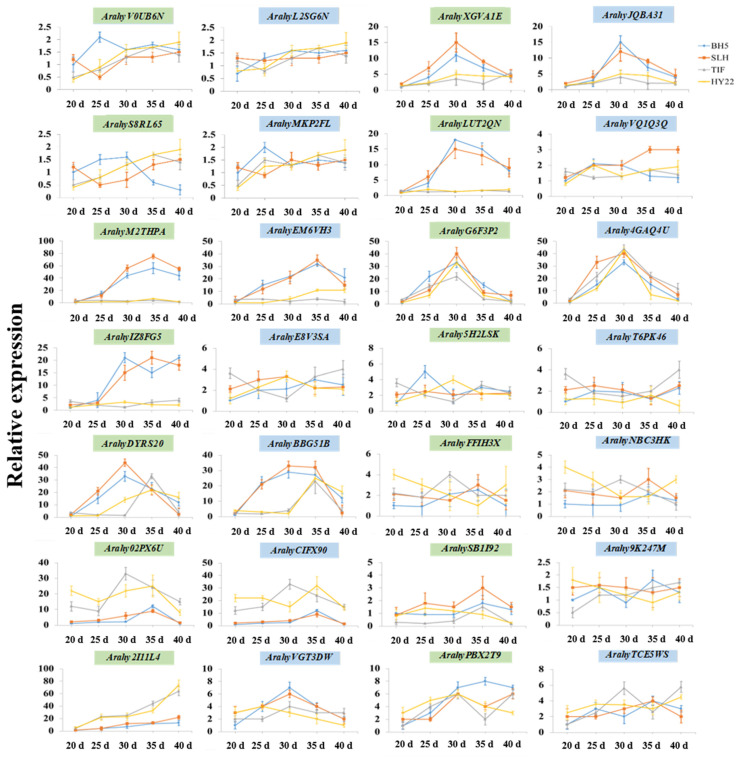
The relative expression patterns of early and late-flowering peanut genotypes. Three biological replicates with their respective three technical replicates were performed.

**Figure 9 ijms-23-12429-f009:**
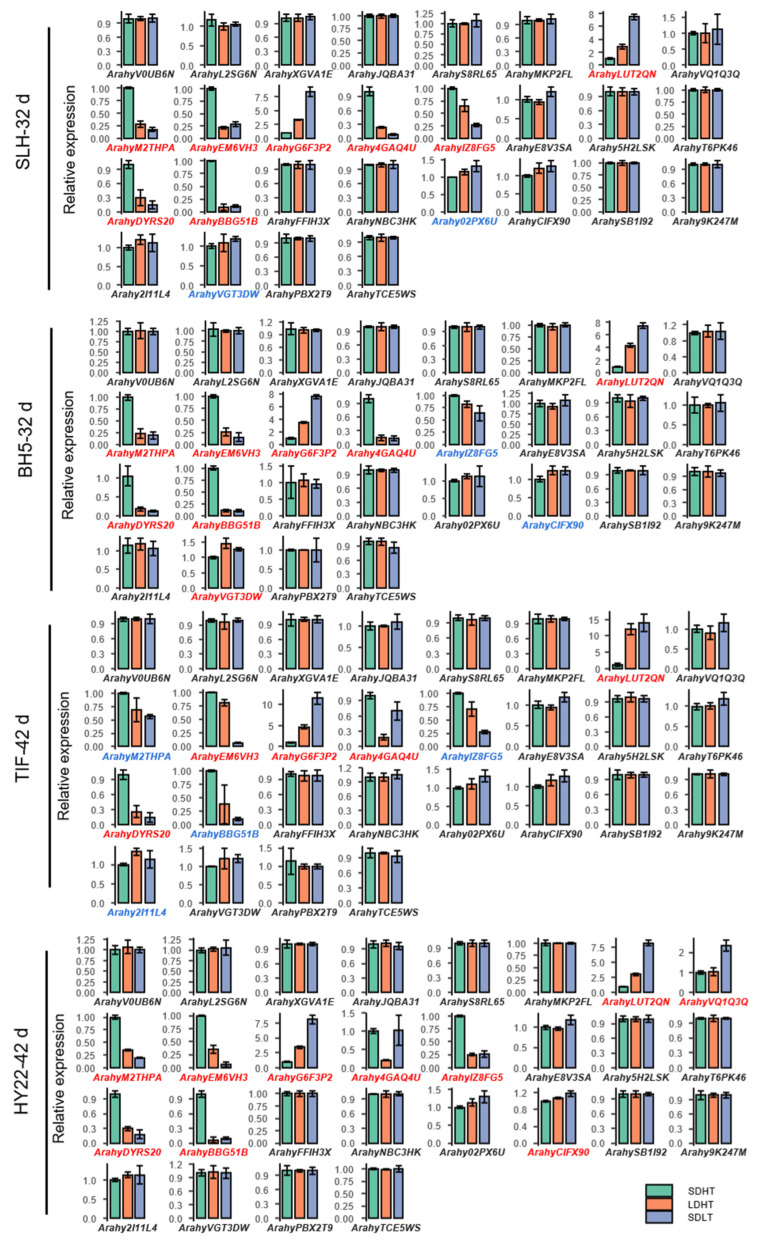
*PEBP* gene expression patterns in different peanut genotypes with different flowering times. The short-day high temperature group (SDHT) was compared with the long-day high temperature group (LDHT) and the short-day low temperature group (SDLT) by Student’s *t*-test. Bars indicated standard error of the mean. The genes marked in red have significant changes in expression level under both long-day and low-temperature conditions, while the genes marked in blue are significantly different in expression level only under low-temperature conditions.

## Data Availability

Not applicable.

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
