# Peer review of "Comparative Genomic and Expression Analysis Insight into Evolutionary Characteristics of PEBP Genes in Cultivated Peanuts and Their Roles in Floral Induction"

_ijms, 2022, doi:10.3390/ijms232012429_

Round 1

Reviewer 1 Report

The authors analyzed the PEBP gene of cultivated peanut, and explored the expression patterns to different genotypes, photoperiods and temperatures, this can laid the foundation for the functional study of PEBP gene in cultivated peanut and the adaptation of peanut to different environments. But there are two serious defects:

1.     The discussion didn't go far enough. Most of the content is a rehash of the results rather than a discussion. The cited references are all soybean literatures. It is suggested that the discussion section should be rewritten to add literature on Arabidopsis, snapdragon, etc.

 2.     Figure 9: It is not appropriate to put the expression levels of different genes into a single graph. Different genes have different primers, and the primers have different amplification efficiency, so the final expression level is different. But the expression level of the same gene in different tissues and species can be compared.

 Minor things need to be paid attentation:

1. Line 126: “A. thalianais better replaced by “Arabidopis”

2. Line 139: duranensis and the tetraploid, the “and” should not be italic.

3. Line 179: “AhFT6A” should be italic.

4. Line 279-280: “such as AhFT1, AhFT6, AhFT8, AhTFL2, AhTFL4 and AhMFT2 showed no significant difference in expression patterns between the early and late flowering genotypes”. Is the difference in expression patterns of AhFT1 between the early and late flowering genotypes not significant? I think the difference is significant. How do you determine the criteria of difference?

5. Line 333: “PEBP proteins in peanut can be divided into three different subfamilies, 333 FT, TFL and MFT.” PEBP, FT, TFL and MFT are protein names, these words should not be italicized.

6. Line 351: “Archis species”, should be “Arachis species”

Author Response

Response to Reviewer 1 Comments

Point 1: The discussion didn't go far enough. Most of the content is a rehash of the results rather than a discussion. The cited references are all soybean literatures. It is suggested that the discussion section should be rewritten to add literature on Arabidopsis, snapdragon, etc.

Response 1: Thank you very much for your suggestion, we have rewritten the discussion part and added other relevant literature.

Point 2: Figure 9: It is not appropriate to put the expression levels of different genes into a single graph. Different genes have different primers, and the primers have different amplification efficiency, so the final expression level is different. But the expression level of the same gene in different tissues and species can be compared.

Response 2: Thank you very much for your valuable suggestion, we have changed the graph to plot each gene separately under SDHT, LDHT and SDLT treatments  (Figure 9).

Point 3: Minor things need to be paid attentation:

  1. Line 126: “A. thaliana” is better replaced by “Arabidopis”

Response 3: Thanks for your suggestion,  has been corrected.

  1. Line 139: “duranensis and the tetraploid”, the “and” should not be italic.

Response 4: Thanks for your suggestion, we have corrected it.

  1. Line 179: “AhFT6A” should be italic.

Response 4: Thanks for your suggestion. In this manuscript, we use the original ID for the genes to facilitate the mapping of PEBP from wild peanuts to PEBP genes from cultivated peanuts. And the text involving genes in the text is italicized.

  1. Line 279-280: “such as AhFT1, AhFT6, AhFT8, AhTFL2, AhTFL4 and AhMFT2 showed no significant difference in expression patterns between the early and late flowering genotypes”. Is the difference in expression patterns of AhFT1 between the early and late flowering genotypes not significant? I think the difference is significant. How do you determine the criteria of difference?

Response 4: Thanks for your suggestion. We redesigned specific primers for each PEBP gene and re-identified the expression pattern of PEBP genes. In the comparison of gene expression patterns between early-flowering and late-flowering genotypes, we considered genes whose expression levels differed by more than 2 fold at a certain time point as significant differences in expression patterns.

  1. Line 333: “PEBP proteins in peanut can be divided into three different subfamilies, 333 FT, TFL and MFT.” PEBP, FT, TFL and MFT are protein names, these words should not be italicized.

Response 5: Thank you for pointing out this error, we have corrected the corresponding position in the text.

  1. Line 351: “Archis species”, should be “Arachis species”

Response 6: Thanks for suggestion. The sentence in which this word is located has been removed in the new manuscript, but we have checked the spelling of all “Arachis species”.

Reviewer 2 Report

The work on the identification and characterization of peanut PEBP genes from cultivated peanut genome seems interesting. However, the written of the manuscript need to be improved, it is almost impossible to properly comprehend the work authors actually wanted to present. Moreover, the identification of PEBP genes from wild and cultivated peanut have been performed by Jin et al., 2019.

1) Line 81 to 84, “In the present study, we performed the first excavation and identification of PEBP in the peanut genome (A. hypogaea)……”. The identification of cultivated peanut (A. hypogaea) PEBP genes have been performed by Jin et al., 2019 (Molecular and transcriptional characterization of phosphatidyl ethanolamine-binding proteins in wild peanuts Arachis duranensis and Arachis ipaensis), and this study is not the first excavation and identification of PEBP in the peanut genome (A. hypogaea).

2) In the manuscript, a total of 26 protein sequences from A. hypogaea, 13 from the A. duranensis, and 14 from the A. ipaensis were identified as PEBP proteins. It is not consistent with Jin et al., 2019. What is the difference between this study and Jin’s work?

3) The subcellular localization of representative PEBP genes in different subgroups is necessary.

4) The expression of A. hypogaea PEBP genes in Figure 6 have been identified by Jin et al., 2019 (Additional file 9).

5) Further, a lot of clarity in writing of the work done is required.

6) Many data about wild and cultivated peanut have been reported by Jin et al., 2019, please confirm in this manuscript.

Author Response

Response to Reviewer 2 Comments

Point 1: Line 81 to 84, “In the present study, we performed the first excavation and identification of PEBP in the peanut genome (A. hypogaea)……”. The identification of cultivated peanut (A. hypogaea) PEBP genes have been performed by Jin et al., 2019 (Molecular and transcriptional characterization of phosphatidyl ethanolamine-binding proteins in wild peanuts Arachis duranensis and Arachis ipaensis), and this study is not the first excavation and identification of PEBP in the peanut genome (A. hypogaea).

Response 1: Thank you very much for your review, it is of great significance to the improvement of our research. For a more accurate presentation, we removed the previous findings by Jin et al. 2019. And we revise the statement in the introduction as follows: “The PEBP genes in diploid wild peanut species were identified, and the PEBP genes in cultivated peanut and tissue-specific expression pattern based on transcriptome data was analyzed. In order to further understand the gene structure and function of PEBP genes in cultivated peanut, in the present study, we re-identified the PEBP genes of cultivated and wild peanut, explored its evolutionary characteristics in cultivated peanut through comparative genomics, and further identified its floral induction respond to genoytpes, photoperiod and temperature.”

Point 2: In the manuscript, a total of 26 protein sequences from A. hypogaea, 13 from the A. duranensis, and 14 from the A. ipaensis were identified as PEBP proteins. It is not consistent with Jin et al., 2019. What is the difference between this study and Jin’s work?

Response 2: Thanks for your question. In this study, we re-identified wild and cultivated peanuts using the same parameters, with 32 wild peanut genes consistent with the findings of Jin et al. 2019. But we identified 33 PEBP genes using the same protocal with wild peanut. Jin et al. 2019 identified 32 PEBP genes in cultivated peanut. ArahyFW8Z6T was included in the results of Jin et al. 2019, while Arahy1WC97K and ArahyT6PK46 were found in this study. It may be due to the different parameters we set to identify PEBP proteins. Since ArahyFW8Z6T contains only 78 amino acid residues, it is much short than the amino acid length of typical PEBP protein and could not be identified by HMM-based methods. Therefore, we excluded ArahyFW8Z6T. Arahy1WC97K and ArahyT6PK46, which was not included in Jin's study, in gene length, isoelectric point and conserved domains all fit the parameters of PEBP protein and were therefore retained (Table S1). Relevant texts we state in the Discussion section

Point 3:  The subcellular localization of representative PEBP genes in different subgroups is necessary.

Response 3: Thanks for your suggestions on our research, which we consider very valuable. We started the work of subcellular localization and planned to select 3 representative PEBP genes to transfer into tobacco leaves, but it will take some time to design primers to amplify the target gene and related molecular biology experiments. At present, we have only successfully identified one FT subgroup gene ArahyEM6VH, whose localization in the cytoplasm was consistent with our predicted results (Table S1). Since the subcellular localization of the representative PEBP genes has not been completed, the results of this part are not included in this paper.

(Please see file attached)

ArahyEM6VH3 subcellular localization results

Point 4: The expression of A. hypogaea PEBP genes in Figure 6 have been identified by Jin et al., 2019 (Additional file 9).

Response 4: Thank you for your reminder, we have deleted this part of the data. Our study of PEBP gene mainly focused on its role in early-flowering and late-flowering genotypes, and changes in the expression pattern of PEBP gene under different photoperiod and temperature treatments. Thus, key genes involved in flowering regulation in cultivated peanut were identified.

Point 5:  Further, a lot of clarity in writing of the work done is required.

Response 5: Thank you very much for your reminder, we have analyzed the data of the entire article, and rewritten the results and discussion parts to better state our work.

Point 6: Many data about wild and cultivated peanut have been reported by Jin et al., 2019, please confirm in this manuscript.

Response 6: Thanks for your valuable suggestiongs. In this study, we used the same method to re-identify the PEBP genes of wild and cultivated peanut. The use of the same parameters facilitated the comparison of the PEBP genes between wild and cultivated peanuts. Although previous study of Jin et al. (2019) identified the gene structure and conserved motif of PEBP gene in wild peanut species, no analysis of the gene structure of cultivated peanut was performed. Additionally, we have removed transcriptome-based analyses of PEBP gene expression in different tissues. We performed a comparison of the A and B subgenomes of cultivated peanut species, collinearity between cultivated peanut and other species, the induction of PEBP genes into flowering in different flowering genotypes, and the identification of PEBP gene responses to photoperiod and temperature expression mode. These results are not included in the results of Jin et al.

Round 2

Reviewer 1 Report

It can be accept in present form

Reviewer 2 Report

The authors have replied all my concerns.